# Radiographic and Histopathological Characteristics of Chronic Nonbacterial Osteomyelitis of the Mandible in Pediatric Patients: Case Series

**DOI:** 10.3390/diagnostics15121549

**Published:** 2025-06-18

**Authors:** Mohammed Barayan, Nagla’a Abdel Wahed, Narmin Helal, Hisham Abbas Komo, Durer Iskanderani, Raghd Alansari, Nada A. Alhindi, Azza F. Alhelo, Hanadi Khalifa, Hanadi Sabban

**Affiliations:** 1Department of Oral Diagnostic Sciences, Faculty of Dentistry, King Abdulaziz University, Jeddah 21589, Saudi Arabia; 2Department of Pediatric Dentistry, Faculty of Dentistry, King Abdulaziz University, Jeddah 21589, Saudi Arabia; 3Oral and Maxillofacial Surgery, King Abdulaziz University Dental Hospital, Jeddah 21589, Saudi Arabia; 4Radiology Department, King Abdulaziz University Dental Hospital, Jeddah 21589, Saudi Arabia

**Keywords:** autoinflammatory bone disorder, chronic nonbacterial osteomyelitis, mandibular expansion, mixed bone lesion, periostitis

## Abstract

**Background and Clinical Significance:** Chronic nonbacterial osteomyelitis (CNO) of the jaw is a rare autoinflammatory bone disorder that primarily affects children and adolescents. Diagnosing CNO of the mandible can be challenging due to its rarity, and the clinical and radiographic findings overlap with those of other bone disorders. **Case Presentation:** This case series retrospectively presents four female pediatric patients (9–12 years old) diagnosed with mandibular CNO. The patients were treated at King Abdulaziz University Dental Hospital, Jeddah, Saudi Arabia, between 2018 and 2024. Clinical features and radiographic and histopathological findings were evaluated. All cases had mandibular swelling and pain. Radiographic features consistently revealed mixed sclerotic and radiolucent lesions with bone expansion and periosteal reactions. Histopathological findings revealed viable bone interspersed with varying degrees of fibrous tissue. No evidence of bacterial colonies or inflammation was observed. This case series highlights the radiographic and histopathological features of CNO in the mandible of pediatric patients. The mixed radiographic features and variability of histopathological findings combined with the refractory nature of the lesions contribute to diagnostic complexity. Diagnostic challenges include differentiating CNO from other inflammatory and fibro-osseous conditions. The presence of recurrent episodes of pain, the formation of subperiosteal bone, periostitis, lysis of the cortical layer, expansion of the mandibular canal, and sterile bone biopsies with nonspecific inflammatory changes were related mainly to CNO. **Conclusions:** These findings underscore the need for increased awareness and a multidisciplinary approach for accurate diagnosis and management of CNO. Conservative management, particularly in dental cases, avoids prolonged unnecessary use of antibiotics, and the prescription of nonsteroidal anti-inflammatory drugs should be followed.

## 1. Introduction

Chronic nonbacterial osteomyelitis (CNO) of the jaw is a rare autoinflammatory bone disorder that primarily affects children and adolescents, presenting unique diagnostic and therapeutic challenges for dental and medical professionals. Clinically, a single bone may be affected, or multiple lesions may be found, a condition that is termed chronic recurrent multifocal osteomyelitis (CRMO) [1,2,3].

The terms chronic nonbacterial osteomyelitis, chronic multifocal symmetrical osteomyelitis, chronic diffuse sclerosing osteomyelitis, and chronic sclerosing osteitis are most likely different types of the same condition [2,4,5]. Lesions are most commonly found in the metaphysis of the long bones, vertebrae, and clavicles, and involvement of the mandible is rare [6].

Diagnosing CNO of the jaw can be challenging because of its rarity and similarity to other bone disorders. The clinical presentation of CNO in the jaw can be variable and often mimics other conditions, such as bacterial osteomyelitis, fibrous dysplasia, tumors, or other diseases. CNO of the jaws is depicted by persistent or recurrent episodes of bone inflammation without an identifiable infectious etiology. Patients typically present with localized pain, swelling, and tenderness in the affected area of the mandible or maxilla. In some cases, trismus, facial asymmetry, or dental abnormalities may be observed. The age of onset is usually childhood, with reports of cases as young as 3 years old [6].

Radiographic findings may include osteolytic lesions, periosteal reactions, and sclerotic changes [7]. Advanced imaging techniques such as magnetic resonance imaging (MRI) and bone scintigraphy can be helpful in identifying multifocal involvement and assessing disease activity [8].

The differential diagnosis of CNO in the jaw includes bacterial osteomyelitis, fibrous dysplasia, Langerhans cell histiocytosis, and malignant bone tumors [9]. Fibrous dysplasia, in particular, can present similarly to CNO, especially in the mandible. A case report by Zain-Alabdeen et al. in 2022 highlighted the diagnostic dilemma between juvenile fibrous dysplasia and chronic osteomyelitis of the posterior mandible, emphasizing the need for careful clinical, radiographic, and histopathological evaluation [10]. Importantly, microscopic examinations are essential for excluding other diseases. While no specific histopathological features define CNO, findings can range from normal bone to areas of fibrosis, osteolysis, and immune cell infiltration, including neutrophils, monocytes/macrophages, lymphocytes, and plasma cells. However, the absence of bacterial colonies is an important feature in the diagnosis of CNO [7,9,11].

The management of CNO in the jaw remains challenging, with no standardized treatment protocol. Nonsteroidal anti-inflammatory drugs (NSAIDs) are often used as first-line therapies for pain relief and to reduce inflammation. In refractory cases, corticosteroids, bisphosphonates, or disease-modifying antirheumatic drugs (DMARDs) may be considered [12,13].

Disease diagnosis is still challenging for dental clinicians. Therefore, this case series aims to highlight the radiographic and histopathological features of the disease and the management to avoid incorrect primary diagnosis and treatment.

## 2. Case Presentation

This retrospective case series included four patients, aged 9–12 years, who presented to King Abdulaziz University Dental Hospital, Jeddah, Saudi Arabia, between 2018 and 2024 for the evaluation and treatment of CNO. This study was approved by the Research Ethics Committee of the Faculty of Dentistry, King Abdulaziz University (76-03-24 dated 4 June 2024).

### 2.1. Case 1

#### 2.1.1. History and Clinical Examination

An 11-year-old female presented with painful facial swelling in the left mandible, disrupted sleep, and fever, following the extraction of an exfoliating sound from primary tooth #75. Intraoral examination revealed sound teeth on the left side, with pain on percussion for teeth #35 and #36. Both teeth were also nonresponsive to cold and electric pulp testing (EPT).

#### 2.1.2. Radiographic Examination

CBCT imaging (Figure 1) revealed marked buccolingual and superior-inferior expansion of the left mandible, extending from tooth #32 to the distal aspect of #37. The lesion exhibited a mixed sclerotic appearance with osteolytic foci. The borders of the diseased bone were ill-defined and blended with the surrounding healthy bone. Thinning of the cortical borders was noted, along with periosteal reactions and the formation of multiple cortical layers, producing an “onion skin” appearance. Contrast-enhanced MRI revealed osseous changes, including loss of normal fatty marrow signal, bone marrow edema, and periosteal new bone formation. Adjacent buccal and lingual soft tissue swelling was also noted (Figure 2).

#### 2.1.3. Histopathological Examination

An incisional bone biopsy (Figure 3) revealed multiple irregular bone fragments interspersed with abundant fibrous tissue, displaying prominent peritrabecular clefting. Biopsy revealed a lack of inflammation with predominant sclerosis. Notably, no evidence of bacterial colonies or sequestration was observed.

#### 2.1.4. Case Management and Differential Diagnosis

Initial management included a 10-day course of oral antibiotics and NSAIDs, but no clinical improvement was observed. Owing to persistent symptoms, a second course of antibiotics (amoxicillin/clavulanate) was prescribed, yet symptoms progressively worsened after completing the treatment. Over the following months, the patient developed progressive mandibular swelling and pain, leading to a referral to our institute for further evaluation. Owing to the lack of initial improvement, the patient was discharged home on a three-month course of oral antibiotics (clindamycin), as recommended by infectious disease specialists. Close follow-up revealed significant clinical improvement, allowing for the discontinuation of antibiotics upon completion of the prescribed course.

Differential diagnosis based on the histological findings suggested fibro-osseous lesions, particularly fibrous dysplasia (FD). However, the diagnosis of CNO was established based on clinical progression, imaging, and histopathology, alongside the lack of sustained response to antibiotic therapy.

A 7-month follow-up CBCT examination (Figure 4) revealed improvement of the lesion, with slight bone rarefaction. Moderate, uniform expansion of the buccal and lingual plates was observed. The periosteal reaction persisted, with the presence of a double cortex along the buccal and inferior border of the mandible. Eighteen months later, the patient returned with pain related to the left side of the face. The swelling had fully resolved, and pulp vitality had returned in both teeth #35 and #36, as confirmed by positive EPT and cold testing. However, a CBCT scan revealed new buccal cortical discontinuities opposite tooth #36, indicating a recurrent active inflammatory process (Figure 5). The patient will continue to be followed up, but no further treatment is required for these teeth.

### 2.2. Case 2

#### 2.2.1. History and Clinical Examination

A 9-year-old female presented in June 2021 to the emergency room with her mother, complaining of severe, spontaneous pain and swelling on the right side of her face. The episodes of pain and swelling continued for approximately 10 months. Despite the use of antibiotics and multiple dental extractions and treatments over several months, including emergency care for other dental concerns, the swelling persisted. Facial asymmetry, swelling, and enlarged lymph nodes related to the lower right mandible.

#### 2.2.2. Radiographic Examination

Periapical, panoramic, and cone-beam computed tomography (CBCT) imaging (Figure 6 and Figure 7) revealed a lesion extending from the mandibular angle posterior to developing tooth #47 and crossing the midline to the region of tooth #33. The lesion had a mixed patchy appearance, with granular sclerotic areas and radiolucent foci fused together. The mandibular canal was traceable within the lesion, with a location suggestive of its superior displacement. There was a marked enlargement and expansion of the mandible with a thin outer cortex and diffuse borders. The follicular cortex of developing teeth was altered to the lesion bone pattern; however, the presence of a double cortex on the inferior border of the mandible was observed.

#### 2.2.3. Histopathological Examination

A biopsy of the bone (Figure 8) revealed multiple trabeculae of viable bone surrounded by non-inflammatory connective tissue. These trabeculae exhibited osteoblastic rimming in focal areas and a modest number of osteoclasts in the form of multinucleated giant cells. In addition, clefting, which separates the bone from the connective tissue, was evident.

#### 2.2.4. Case Management and Differential Diagnosis

The radiographic bone pattern and the relationship between the lesion and the mandibular canal initially pointed toward a diagnosis of FD. However, the presence of a periosteal reaction, particularly the appearance of a double cortex, along with the histopathological findings, shifted the diagnosis toward CNO. The patient experienced recurrent episodes of pain and persistent swelling that did not improve despite multiple courses of antibiotics and several dental extractions. These clinical features, combined with sterile biopsy results, supported the final diagnosis of CNO. Once confirmed through imaging and histology, the patient was treated conservatively with NSAIDs, which led to gradual improvement over several months. This outcome is consistent with the literature, which supports the use of NSAIDs as the first-line therapy for CNO [10].

### 2.3. Case 3

#### 2.3.1. History and Clinical Examination

A 10-year-old female presented in May 2024 with pain and swelling on the right side of the face, which persisted for four months. The swelling extended from the anterior to the posterior right mandible, with no lymph node involvement noted. An aspiration biopsy revealed blood flow, prompting further investigation. The case was evaluated by a pediatric dentist to rule out inflammatory causes related to carious teeth.

#### 2.3.2. Radiographic Examination

Panoramic and CBCT imaging (Figure 9 and Figure 10) revealed a lesion involving the right mandible, extending from tooth #34 to #47. The mixed lesion spanned from the alveolar crest to the inferior border of the mandible, with multiple layers of linear periosteal new bone formation along the buccal and lingual cortices. It showed a mixed, patchy appearance with granular sclerotic areas fused to radiolucent foci (“moth-eaten” pattern). There was a buccolingual and inferior expansion of the bone. Furthermore, a contrast-enhanced CT scan revealed periostitis, sclerosis, and expansion with tiny areas of cortical erosion, strongly suggestive of chronic osteomyelitis of the right mandible.

#### 2.3.3. Histopathological Examination

Incisional bone biopsy revealed sterile, normal, viable bone, confirming the non-infectious nature of the lesion (Figure 11).

#### 2.3.4. Case Management and Differential Diagnosis

Following clinical and radiographic evaluation, empirical antibiotics were initiated in conjunction with NSAIDs and paracetamol to manage pain and address the possibility of low-grade bacterial osteomyelitis. However, the absence of clinical improvement and the sterile biopsy findings prompted reconsideration of the diagnosis. The persistence of symptoms despite antibiotic therapy, along with the presence of layered periosteal new bone formation, cortical expansion, and sclerosis observed on contrast-enhanced CT, was strongly suggestive of CNO. The radiology report further supported this by noting features consistent with chronic recurrent multifocal osteomyelitis and recommending systemic skeletal evaluation. These findings align with previous literature emphasizing the role of advanced imaging and histopathology in distinguishing CNO from bacterial osteomyelitis and malignancy [10]. After confirming the diagnosis, the focus shifted to conservative management with NSAIDs alone, which led to notable clinical improvement over time. Escalation to corticosteroids or disease-modifying agents was not necessary. The case underscores the importance of early recognition of CNO to avoid unnecessary antibiotic exposure and highlights the diagnostic value of advanced imaging in guiding clinical decisions.

### 2.4. Case 4

#### 2.4.1. History and Clinical Examination

A 12-year-old female initially presented to the emergency clinic in December 2018 with swelling associated with tooth #75, which was subsequently extracted. Antibiotics, as well as paracetamol and NSAIDs, were administered. One year later, she returned with recurrent pain and swelling, prompting further investigation.

#### 2.4.2. Radiographic Examination

CBCT and medical CT imaging (Figure 12) showed buccolingual expansion with a prominent inflammatory periosteal reaction extending from the distal region of tooth #37 along the buccal aspect and crossing the midline to the level of tooth #43. The lesion had an irregular outline with a patchy, mixed appearance of sclerosis and osteolytic foci. Minimal soft tissue swelling was noted on medical CT, with no evidence of lymphadenopathy. Additionally, FLOW and SPECT-CT examinations revealed abnormal uptake in the left mandible with no evidence of hyperemia and no other significant bony abnormalities throughout the skeleton.

#### 2.4.3. Histopathological Examination

The biopsy revealed viable bone with no bacterial colonies.

#### 2.4.4. Case Management and Differential Diagnosis

Initially, the patient received empirical antibiotics, paracetamol, and NSAIDs to manage the acute swelling associated with tooth #75. This management approach aligns with the clinical course described in the literature, where patients often undergo empirical antibiotic therapy prior to receiving a diagnosis of CNO. Although her symptoms subsided temporarily, the recurrence of mandibular pain and swelling one year later prompted further imaging and biopsy. Radiographic findings, including mixed sclerotic and lytic changes, buccolingual expansion, and periosteal reaction, raised suspicion of a chronic inflammatory condition. The radiographic and histological features observed were consistent with those previously described in mandibular CNO cases. The absence of systemic signs, lymphadenopathy, or hyperemia, along with a sterile biopsy confirming viable bone without microbial colonies, led to a final diagnosis of chronic nonbacterial osteomyelitis (CNO). Conservative management with NSAIDs was continued, and no additional treatment was required at that stage. The patient showed gradual clinical improvement, which was consistent with the favorable response to NSAIDs commonly reported in the literature [10].

Table 1 summarizes the clinical, radiographic, and histologic features and management of the four cases.

## 3. Discussion

This case series highlights the complexity of diagnosing mandibular lesions in pediatric patients. All four cases involved young females aged 9–12 years, and the clinical and radiographic presentations were consistent with the hallmark features of CNO: swelling, pain, and a mixed radiographic appearance combining sclerotic and radiolucent components. The presence of periosteal reactions, such as a double cortex and onion-skin appearance, is a common radiographic finding, emphasizing the chronic and recurrent nature of the condition. This is in accordance with the findings of Buch et al., who reported that CT imaging reveals intra-bony sclerotic or osteolytic lesions alongside periosteal and soft tissue involvement [2].

CNO should be differentiated from chronic bacterial osteomyelitis, which is characterized by microbial colonization, inflammation-induced bone necrosis, and possible sequestration. Systemic conditions such as diabetes, anemia, or prior radiation exposure, along with bone diseases such as osteoporosis and FD, are recognized as risk factors for chronic bacterial osteomyelitis. The differential diagnosis of CNO and FD is particularly challenging because of overlapping imaging features [14,15]. Misdiagnosis rates of up to 34.6% have been reported [16].

In our study, case 2 presented radiographic features suggestive of FD, including cortical changes, mandibular canal location, and mandibular expansion. This mirrors earlier comparisons between diffuse sclerosing osteomyelitis (an alternative term for CNO) and FD, both of which predominantly affect young females and are typically present unilaterally. However, pain and periosteal bone formation are more characteristic of CNO. FD is more often associated with well-demarcated cortical layers, tooth displacement, and mandibular canal alterations. Moreover, features such as periosteal thickening and fistula formation point toward osteomyelitis rather than FD [10,17,18].

Histopathological variability further complicates the diagnostic process. Case 1 exhibited trabecular bone with fibrous stroma and peri-trabecular clefting, which are features often associated with FD. However, the absence of the “Chinese character” trabecular pattern and the presence of viable bone, without dysplastic changes, favored a diagnosis of CNO, which is consistent with findings described by Jia et al. [5]. FD is a developmental bone disorder linked to GNAS mutations, whereas CNO is an autoinflammatory condition characterized by episodic exacerbations, reinforcing the importance of accurate multidisciplinary evaluation [19].

The histopathological findings in this study align with previous reports of the variable microscopic features of CNO [2,3]. Cases 3 and 4 presented with viable bone with minimal fibrous tissue. Case 1 showed woven bone with prominent peri-trabecular clefting. No bacterial colonies, necrosis, or acute inflammation were found in any patient, which helps distinguish CNO from bacterial osteomyelitis. In case 2, focal osteoblastic rimming and multinucleated giant cells were observed, reflecting a reactive bone process further supporting an auto-inflammatory origin. These findings mirror previously reported sterile bone biopsies with nonspecific inflammatory changes in CNO [20].

All patients in our series were initially managed with empirical antibiotics due to clinical signs such as fever, facial swelling, and soft tissue involvement, which are also common in bacterial osteomyelitis. Case 1, for example, presented with pain, fever, and persistent swelling after the extraction of an exfoliating, sound deciduous tooth. Case 2 had undergone extractions of multiple teeth at the early stages of her management, reflecting how diagnostic ambiguity may lead clinicians to overtreat what appears to be an opportunistic infection. In hindsight, the necessity of dental extractions remains unclear, as no definitive evidence of underlying dental pathology was documented, suggesting that they may have been avoidable in the context of a misinterpreted inflammatory presentation. This underscores the importance of early consideration of CNO in atypical mandibular presentations to avoid irreversible or excessive treatment.

Antibiotic courses in these cases included amoxicillin/clavulanate and clindamycin but yielded no sustained improvement. In case 1, the patient was discharged on a three-month oral antibiotic course based on an infectious disease consultation, largely due to recurrent fever and systemic manifestations that resembled infection, despite the absence of identifiable bacterial organisms on culture or histology. Clinical improvement followed, but it is likely attributed to the self-limiting course of CNO rather than antibiotic efficacy. This experience reflects a well-documented pattern in the literature where antibiotics are often overprescribed before CNO is correctly identified [7,21]. When symptoms fail to resolve, and cultures confirm sterile biopsies, advanced imaging and histological analysis are critical to establish a diagnosis. CBCT and contrast-enhanced CT scans revealed hallmark features of CNO, including periosteal bone layering, sclerosis, and buccolingual expansion [6,13].

Once bacterial and fungal infections were definitively excluded, all patients were treated conservatively with NSAIDs. NSAIDs are the first-line pharmacologic treatment for CNO and have improved pain in 40% to 80% of pediatric patients [12,13]. In our series, NSAID therapy resulted in clinical resolution in all four cases, and no patient required escalation to corticosteroids or methotrexate. This contrasts with reports of systemic or multifocal CNO, where escalation is more frequently needed [4,22]. The mandibular-only localization in our series likely contributed to this favorable response.

In case 1, an 18-month follow-up revealed complete resolution of swelling and pain, along with recovery of pulp vitality in the previously affected teeth, as confirmed by positive cold and EPT tests. This highlights the importance of avoiding premature endodontic or surgical interventions in CNO. Maintaining dental vitality is achievable when the inflammatory process subsides, and unnecessary treatment may be avoided.

For recurrences, our protocol involved repeated imaging and laboratory evaluation to exclude infection or new disease progression. Case 1 showed radiographic evidence of recurrence with cortical irregularities, but the clinical symptoms remained minimal, and no further intervention was needed. In such instances, switching the NSAID type or extending the duration of treatment was sufficient to maintain disease control. This is consistent with the relapsing-remitting nature of CNO [22]. Similar cases in the literature have highlighted that CNO follows a relapsing-remitting course, with periods of spontaneous improvement even in the absence of targeted therapy [23,24].

Hyperbaric oxygen therapy (HBOT) has been considered for chronic infectious osteomyelitis because of its ability to improve oxygenation and tissue healing. However, evidence supporting its use in nonbacterial osteomyelitis remains limited. While a few reports suggest a possible benefit in refractory infectious osteomyelitis, its application in auto-inflammatory bone disorders such as CNO is not supported by high-level evidence [25,26]. Consequently, HBOT was not utilized in our cohort. It may be discussed on a case-by-case basis in multidisciplinary settings, but current guidance does not recommend it for isolated CNO of the mandible.

## 4. Conclusions

This case series highlights the radiographic and histopathological features of CNO of the mandible in pediatric patients. The mixed radiographic appearance, variable histopathological features, and recurrent episodes of pain combined with the refractory nature of the lesions underscore the importance of a multidisciplinary approach in diagnosing and managing this condition. Further research is needed to better understand the pathophysiology of CNO and to develop standardized treatment protocols, particularly in pediatric populations.

## Figures and Tables

**Figure 1 diagnostics-15-01549-f001:**
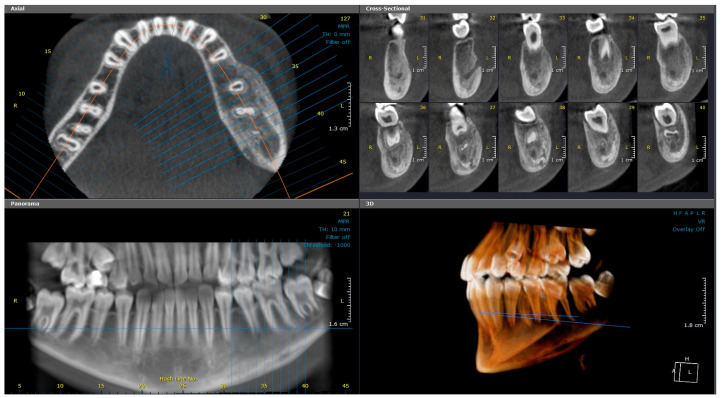
(Case 1) Axial, cross-sectional, reconstructed panoramic, and 3D volumetric images showing the mixed bony lesion of the left mandible with marked buccolingual expansion and cortical layering of the inferior border. The colored lines represent the levels of the cut sections in relation to each other.

**Figure 2 diagnostics-15-01549-f002:**
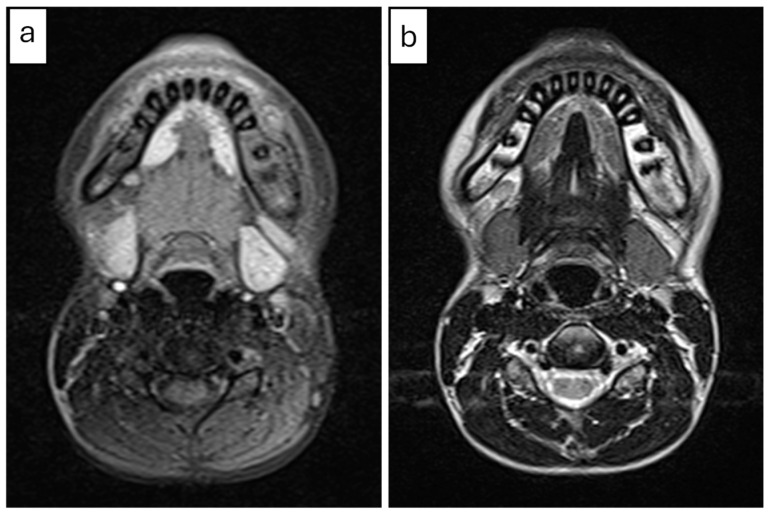
(Case 1) Axial contrast-enhanced MRI images of the mandible: (**a**) T1-weighted and (**b**) T2-weighted images at the same level demonstrate osseous changes in the left mandible, including loss of normal fatty marrow signal, bone edema, and periosteal new bone formation. Adjacent buccal and lingual soft tissue swelling was also noted.

**Figure 3 diagnostics-15-01549-f003:**
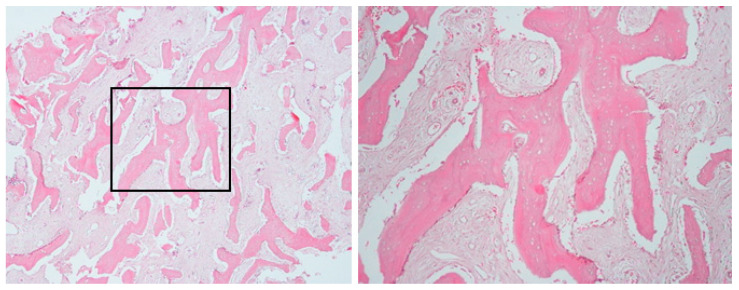
Bone biopsy of case 1, showing irregular bone fragments interspersed with abundant fibrous tissue and prominent peritrabecular clefting. No evidence of bacterial colonies or inflammation was observed.

**Figure 4 diagnostics-15-01549-f004:**
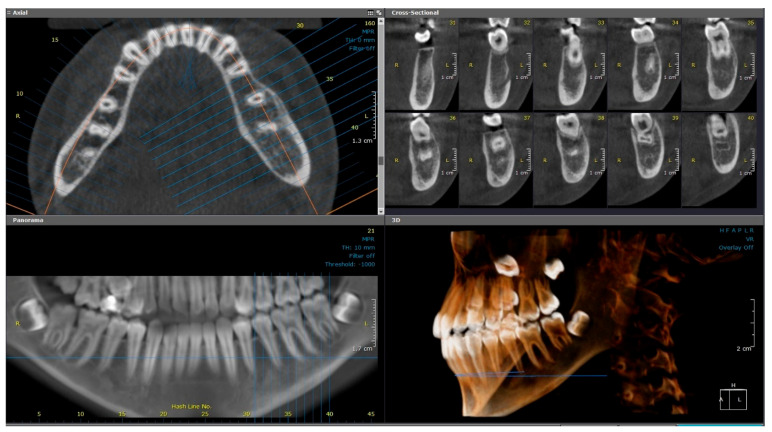
(Case 1—7-month follow-up) Axial, cross-sectional, reconstructed panoramic and 3D volumetric images showing slight bone rarefaction, moderate, uniform buccolingual expansion, and a double cortex at the inferior border of the left side of mandible. The colored lines represent the levels of the cut sections in relation to each other.

**Figure 5 diagnostics-15-01549-f005:**
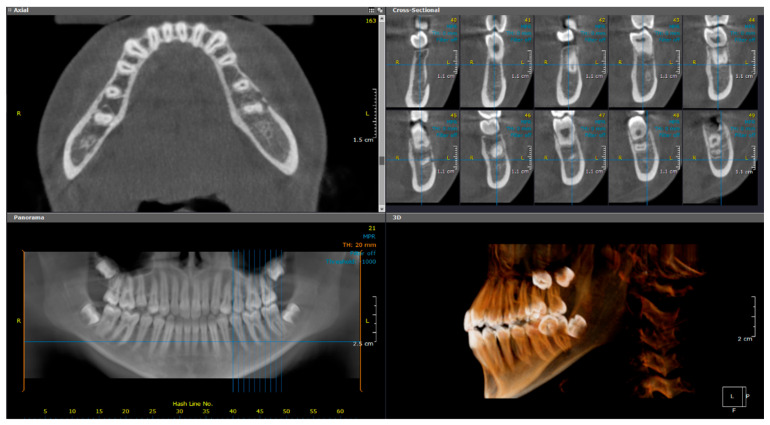
(Case 1—18-month follow-up) Axial, cross-sectional, reconstructed panoramic, and 3D volumetric images showing discontinuities in the buccal cortical outline of the left side of mandible. The colored lines represent the levels of the cut sections in relation to each other.

**Figure 6 diagnostics-15-01549-f006:**
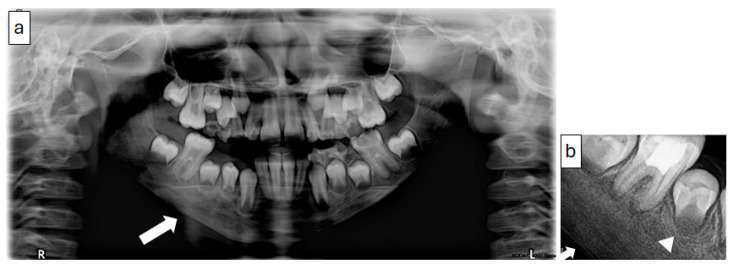
(Case 2) Panoramic (**a**) and periapical (**b**) images showing a mixed patchy bone pattern on the right side of the mandible. Note the double cortex at the inferior border of the mandible (white block arrow) and the loss of the follicular cortex (arrowhead).

**Figure 7 diagnostics-15-01549-f007:**
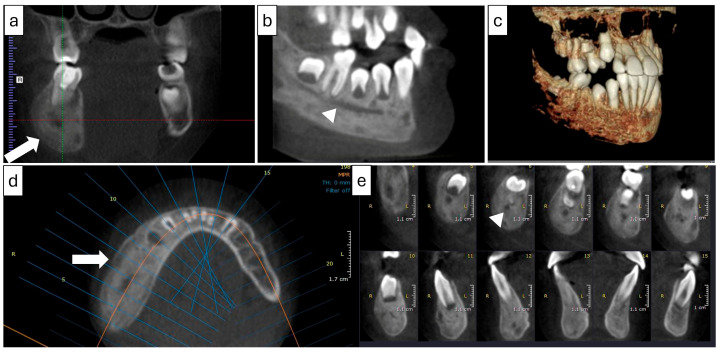
(Case 2) Coronal (**a**), corrected sagittal (**b**), 3D volumetric (**c**), axial (**d**), and series of cross-sectional (**e**) images showing expansile mixed lesions on the right side of the mandible. Note the double cortex at the inferior border of the mandible (white block arrow) and the position of the mandibular canal (arrowhead). The colored lines represent the levels of the cut sections in relation to each other.

**Figure 8 diagnostics-15-01549-f008:**
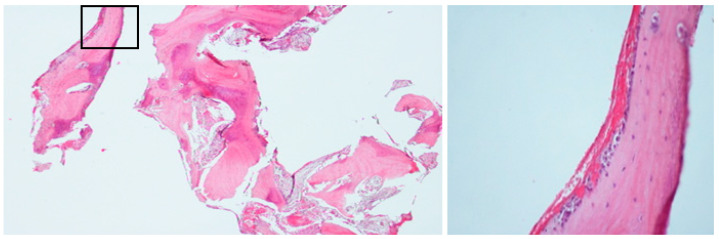
(Case 2) Bone biopsy showing viable trabeculae surrounded by non-inflamed connective tissue with distinct clefting. No evidence of bacterial colonies was observed. Focal areas exhibit osteoblastic rimming and multinucleated giant cells (insert).

**Figure 9 diagnostics-15-01549-f009:**
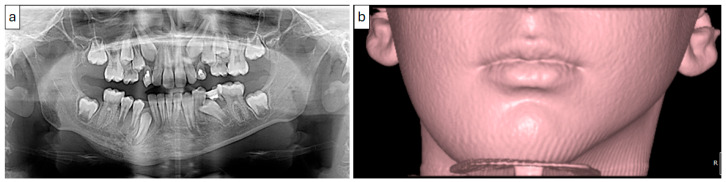
(Case 3) Panoramic image (**a**) showing an expansile mixed bony lesion on the right side of the mandible. Multiple layers of linear periosteal new bone formed at the inferior border of the mandible. The 3D volumetric soft tissue rendering image (**b**) shows facial swelling on the right side.

**Figure 10 diagnostics-15-01549-f010:**
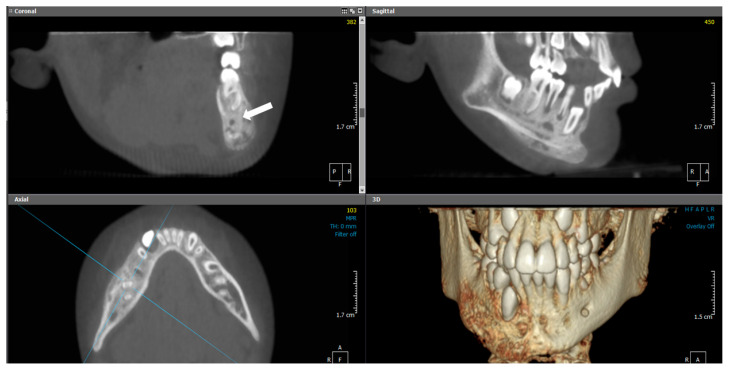
(Case 3) MPR and 3D volumetric images showing extensive mixed lytic and sclerotic areas with periosteal bone reactions. Note the presence of the original cortical boundary of the mandibular bone (white block arrow). The blue lines represent the levels of the cut sections in relation to each other.

**Figure 11 diagnostics-15-01549-f011:**
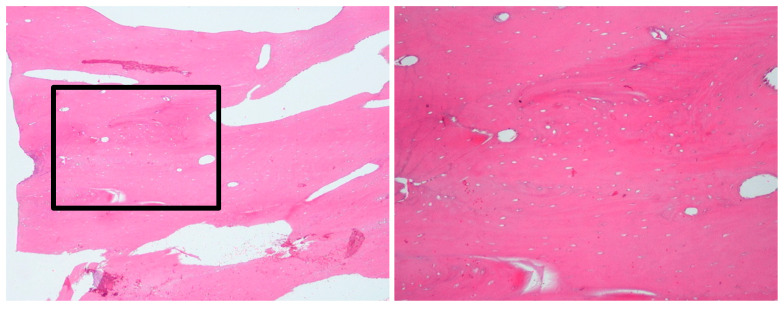
(Case 3) The bone biopsy results revealed normal bone.

**Figure 12 diagnostics-15-01549-f012:**
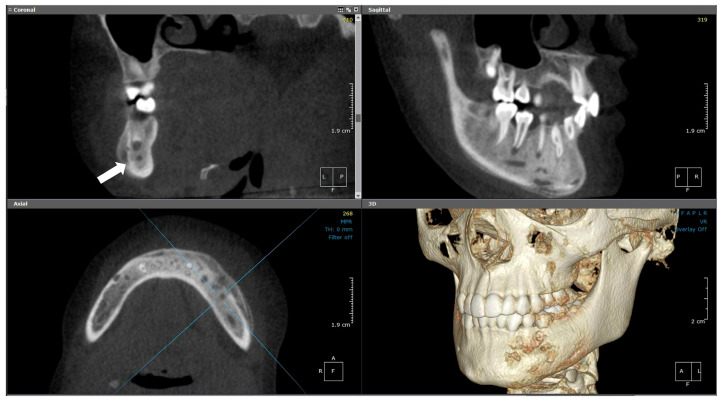
(Case 4) MPR and 3D volumetric images showing the extensive mixed lytic and sclerotic areas extending from tooth #43 to distal #37. Note the periosteal bone reaction and the presence of the original cortical boundary of the mandibular bone (white block arrow). The blue lines represent the levels of the cut sections in relation to each other.

**Table 1 diagnostics-15-01549-t001:** Summary of clinical, radiographic, and histologic features and management of cases.

Case	Age	Lesion Location	Radiographic Appearance	Histopathology	Initial Management	Definitive Diagnosis	Final Treatment	Outcome
1	11	Left mandible (#32–#37)	Mixed sclerotic and lytic lesion with buccolingual expansion and “onion-skin” periosteal reaction	Irregular bone and fibrous tissue, sclerosis, no inflammation or bacteria	Long-term antibiotic course, NSAIDs	CNO confirmed by imaging, histology, and initial failure to respond to antibiotics	Conservative; antibiotics discontinued after improvement	Improved at 7 months; recurrence at 18 months; no further treatment needed
2	9	Right mandible (#33–#47, crossing midline)	Patchy sclerotic and radiolucent lesion, mandibular expansion, double cortex	Viable bone with osteoblastic rimming, no bacterial colonies	Multiple antibiotics, extractions, NSAIDs	CNO based on imaging, histology, and clinical course	NSAIDs only; progressive improvement	Gradual resolution with NSAIDs
3	10	Right mandible (#34–#47, crossing midline)	Granular sclerotic and radiolucent lesion, moth-eaten pattern, layered periosteal reaction	Normal viable bone, sterile	Empirical antibiotics, paracetamol, NSAIDs	CNO confirmed by imaging, CT, and sterile biopsy	NSAIDs only; no need for further escalation	Progressive improvement; under observation
4	12	Left mandible (#37–#43, crossing midline)	Irregular mixed sclerotic/lytic lesion, periosteal reaction, buccolingual expansion	Viable bone with no bacterial colonies	Empirical antibiotics, paracetamol, NSAIDs	CNO confirmed based on recurrence, imaging, and histology	NSAIDs only; no additional therapy needed	Gradual clinical improvement; no recurrence noted

## Data Availability

The original contributions presented in this study are included in the article. Further inquiries can be directed to the corresponding authors.

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
