# Peer review of "Radiographic and Histopathological Characteristics of Chronic Nonbacterial Osteomyelitis of the Mandible in Pediatric Patients: Case Series"

_diagnostics, 2025, doi:10.3390/diagnostics15121549_

Round 1
Reviewer 1 Report
Comments and Suggestions for Authors
Generally a well written report of 4 cases of CMO. The highlight of the report was the difficulties in diagnosis such condition as it should be. The cases management/outcome however seems not complete. For case 2, 3 and 4, it seems once diagnosis is achieved the management outcome are not described in depth. It is important for this disease to be described fully as the treatment responses is also a validation of the diagnosis. This should be added. Otherwise I would also suggest the authors to add a table combining the results such as mean/range age, location of lesion, size of lesion, management, and so on to make it easier for readers to visualized the summary of the cases and make comparison
Author Response
Comment 1: The cases management/outcome however seems not complete. For case 2, 3 and 4, it seems once diagnosis is achieved the management outcome are not described in depth. It is important for this disease to be described fully as the treatment responses is also a validation of the diagnosis. This should be added.
Response 1: Thank you for pointing this out. We agree with your suggestion. Therefore, we have added a paragraph in Section 2.2.4. (line 198-207) describing the management of Case 2. We also have added a paragraph in Section 2.3.4. (line 241-256) describing the management of Case 3. We also have added a paragraph in Section 2.4.4. (line 285-299) describing the management of Case 4.
Comment 2: Otherwise I would also suggest the authors to add a table combining the results such as mean/range age, location of lesion, size of lesion, management, and so on to make it easier for readers to visualized the summary of the cases and make comparison.
Response 2: Thank you for this valuable comment. We agree with your suggestion. We have, accordingly, added table 1 in the text before the discussion section to summarize the cases.
Reviewer 2 Report
Comments and Suggestions for Authors
The paper is interesting, important, and informative. It highlights the diagnostic difficulties and the rare occurrence of the disease. The article demonstrates that establishing the correct diagnosis can take a long time. Most patients receive ineffective antibiotic treatment before the diagnosis of chronic non-bacterial osteomyelitis is reached.
The authors present various diagnostic options. Above all, characteristic anamnesis data are important: recurrent pain and swelling. CT and MRI scans show mixed sclerotic and radiolucent lesions. The biopsy results are also characteristic: no signs of bacterial inflammation are found, only viable bone interspersed with varying degrees of fibrous tissue.
The descriptions of radiographic and histopathologic findings are excellent in all four cases.
- One drawback is that the authors write very little about therapy.
- It is unclear what results the "non-steroidal anti-inflammatory drug" treatment achieved in the patients described in the paper.
- They also do not mention what options are available if this treatment proves ineffective. Some prefer steroid therapy in such cases.
- What is the authors’ opinion on high-pressure oxygen therapy?
I recommend the paper for publication after a minor revision. With its help, the lengthy diagnostic process for some patients may be shortened.
Author Response
Comment 1: One drawback is that the authors write very little about therapy. It is unclear what results the "non-steroidal anti-inflammatory drug" treatment achieved in the patients described in the paper.
Response 1: Thank you for pointing this out. We agree with this comment. Therefore, we have added detailed management descriptions for cases 2, 3, and 4 in Sections 2.2.4. (line 198-207), 2.3.4. (line 241-256), and 2.4.4. (line 285-299), respectively. We also added table 1 to summarize the cases.
Comment 2: They also do not mention what options are available if this treatment proves ineffective. Some prefer steroid therapy in such cases.
Response 2: Thank you for pointing this out. We agree with this comment. We have, accordingly, added in the discussion the effectiveness of NSAIDs as the first-line pharmacologic treatment for CNO. This can be found in line 371-377 and in a paragraph discussing the recurrence protocol (line 383-390).
Comment 3: What is the authors’ opinion on high-pressure oxygen therapy?
Response 3: Thank you for this valuable suggestion. We have, accordingly, modified the discussion section and added a paragraph to do discuss the use of hyperbaric oxygen (line 391-398).
Round 2
Reviewer 1 Report
Comments and Suggestions for Authors
The authors have address all raised concerns well. The paper now reads well and fully describe the case progress and outcome. No further comments from this reviewer